# Effect of Ultraviolet—A Radiation on Alicyclic Epoxy Resin and Silicone Rubber Used for Insulators

**DOI:** 10.3390/polym14224889

**Published:** 2022-11-12

**Authors:** Xiaoqing Wang, Haonan Fan, Wenrong Li, Yuyang Zhang, Ruiqi Shang, Fanghui Yin, Liming Wang

**Affiliations:** 1Maintenance and Test Center of Extra High Voltage Power Transmission Company of China Southern Power Grid, Guangzhou 510663, China; 2Chengnan Power Supply Branch, State Grid Tianjin Electric Power Company, Tianjin 300201, China; 3Laboratory of Advanced Technology of Electrical Engineering and Energy, Shenzhen International Graduate School, Tsinghua University, Shenzhen 518055, China

**Keywords:** alicyclic epoxy resin, hygrothermal environment, silicone rubber, ultraviolet-A radiation

## Abstract

Compared with the high-temperature vulcanized silicone rubber (HTVSR) insulator, the alicyclic epoxy resin insulator has higher hardness and better bonding between the core and the sheath. This makes the latter very promising in the coastal area of Southern China. Outdoor insulators are often subjected to high intensity of ultraviolet (UV)-A radiation. The influence of UV-A radiation is significant for alicyclic epoxy resin insulators. To help address the concern, the surface of two kinds of samples, namely the alicyclic epoxy resin insulator and HTVSR insulator, with UV-A aging time was characterized by tests of scanning electron microscope (SEM), X-ray photoelectron spectroscopy (XPS), and Fourier transform infrared spectroscopy (FTIR). The operation properties (mechanical properties, hydrophobicity) for outdoor insulators were also analyzed. It was found that the appearance color of the alicyclic epoxy resin has changed greatly, and there is a certain degree of fading. The mechanical properties of the alicyclic epoxy resin are maintained well and, the hydrophobicity decreases gradually. For silicone rubber, the appearance color change of silicone rubber is smaller, and the mechanical properties of silicone rubber decreased greatly. In addition, although the hydrophobicity of silicone rubber decreased gradually, it is still better than that of alicyclic epoxy resin. Both materials have broken chemical bonds, but the degree is relatively light, which meets the requirements of insulators for outdoor operation.

## 1. Introduction

As an indispensable part of power transmission lines, insulators play an important role in mechanical support and electrical insulation. China began to develop composite insulators in the 1980s. Under the background of large-scale pollution flashover in the 1990s, composite insulators were widely used in China’s power transmission system for their excellent anti-pollution performance and easy maintenance [1,2,3]. Since then, high-temperature vulcanized silicone rubber (HTVSR) insulators have been used for more than 40 years [4,5,6]. However, silicone rubber, influenced by the inorganic fillers and its characteristic, is soft and vulnerable to external forces such as strong wind and birds peck, because of its characteristics of large molecular spacing and weak molecular inter-atomic forces [7,8,9,10]. When the mandrel is exposed to air, it suffers from the action of light, moisture, pollution, and mandrel mechanical performance degradation. In that case, it could lead to malignant accidents such as fracture of insulator string [11,12]. In order to solve these problems, some researchers brought forward a solution by replacing the insulating material of silicone rubber with alicyclic epoxy resin.

Insulators run outdoors all around the year. Long-term exposure to ultraviolet light will lead to photooxidation degradation of molecular materials. After aging, the mechanical strength, surface hydrophobicity, and other macroscopic properties of insulating materials may change obviously. Ultraviolet light can be divided into ultraviolet (UV)-A, UV-B, and UV-C according to its wavelength. UVA stands for long wavelength in ultraviolet light with a wavelength between 400 nm and 315 nm. Most UVB and almost all UVC shortwave UV light are blocked by the atmosphere. However, due to the wavelength, 98% of UVA can pass through the earth’s atmosphere and reach the ground. Therefore, most of the UV light that reaches the earth’s surface is UVA [13].

Li carried out UV-A aging of silicone rubber materials for 1000 h [14]. The test result showed that with the increase of UV aging time, the tensile strength, elongation at break, volume resistivity, and surface resistivity of silicone rubber materials decreased while the hardness gradually increased. In addition, the breakdown strength first decreased and then increased. Gao compared the UV aging resistance of modified alicyclic epoxy material and silicone rubber material and found that although the UV aging resistance of modified alicyclic epoxy resin is slightly lower than that of silicone rubber materials, it still meets the requirements of outdoor insulation [15].

The ultraviolet aging mechanism of organic insulating materials has been studied in many works of the literature [16,17,18,19]. Under the action of UV aging, the chain segment fracture degradation process of organic insulating materials may occur. In this process, photooxidation and photodegradation reactions will cut the chemical bonds with low bond energy in organic insulating materials. Because of this reaction, the generated free radicals will undergo further oxidation and a certain degree of re-crosslinking reaction will occur between the severed chemical bonds. As a new type of insulator, the aging performance of alicyclic epoxy insulators needs to be compared with silicone rubber insulators to test their reliability after the operation.

Therefore, the UV aging resistance of alicyclic epoxy resin materials is studied in this paper. The alicyclic epoxy resin materials are exposed to UVA under the UV lamp and then taken out after different aging times. The tensile strength, tensile elongation, hydrophobic, scanning electron microscopy (SEM), Fourier infrared spectrum, and X-ray photoelectron spectroscopy were conducted to study the aging mechanism. Meanwhile, its aging resistance performance was compared with silicone rubber insulating material. 

## 2. Experimental Works

### 2.1. Materials

The alicyclic epoxy resin samples used in this paper are made by an automatic pressure gel molding process by a manufacturer [20,21,22]. The formula and process of the test sample are the same as that of the 110 kV alicyclic epoxy insulator sheath produced by the manufacturer, and the HTVSR sample is produced by Dalian Insulator Group Co., Ltd. (Dalian, China). The basic chemical structures of the materials studied are shown in Figure 1.

### 2.2. Methods

#### 2.2.1. UV-A Aging Process 

In this paper, the UV-A aging test was carried out with a UV aging test chamber produced by Esry Instrument Technology Co., LTD (Guangdong, China), as shown in Figure 2. The xenon lamp emitted a continuous spectrum from short-wave UV to infrared radiation. In order to simulate natural UV radiation reaching the earth’s surface, the short-wave UV radiation below 300 nm of light was blocked to generate the UV radiation mainly composed of UV-A radiation [12,23]. The radiation distance between the radiation source and the specimens is 20 cm. The infrared light with low photon energy can heat the sample surface by radiation although the bonds in silicone rubber will not be broken. Therefore, the temperature was controlled by regulating the airflow of a fan. During the experiment, UV irradiance was set to 0.76 W/m^2^. Moreover, the exposure period was set to 8 h dry irradiation and 4 h dark condensation. In these periods, the temperature was controlled at 60 ℃ during irradiation and 50 ℃ during condensation. The distance between the test sample and the lamp is 50 ± 3 mm. The surfaces of epoxy resin and HTVSR samples with different shapes are wiped clean with anhydrous ethanol and non-woven cloth. After that, the samples were placed on the test rack of the UV aging chamber for the UV aging test. The aging time was 0 h, 120 h, 240 h, 600 h, and 1000 h, respectively. In Guangdong, China, the natural solar irradiation is about 4234.62 MJ/m^2^, of which the ultraviolet light that can reach the ground accounts for about 1%. The wavelength distribution of type 1A (UVA–340) fluorescent ULTRAVIOLET lamp in standard GB/T 16422.3—2014 is as follows: 290 ≥ *λ* ≥ 320 accounts for 5.4%, 320 > *λ* ≥ 360 accounts for 38.2%, 360 > *λ* ≥ 400 accounts for 56.4%. Based on these data, it can be calculated that 1000 h of UV aging is equivalent to 4 years of natural UV irradiation in Guangdong.

#### 2.2.2. Scanning Electron Microscope 

To compare the surface change after UV-A radiation, the microtopography of specimens was imaged by SEM (JSM–6460, JEOL, Peking, China) at 500× magnifications with an applied voltage of 5 kV. Since both HTV and alicyclic epoxy resin are insulating materials, they were sputter-coated with gold before the experiment.

#### 2.2.3. Mechanical Characterization 

In this study, tensile properties were tested according to ISO 37:2017 with an electronic tension test machine made by Machine Equipment Co., Ltd., (Shanghai, China) [24,25,26]. In accordance with ISO 37:2017, the samples were dumbbell-shaped with a thickness of 2 mm and a test length of 25 mm. The experiment was conducted with a tensile speed of 500 mm/min. Five samples of each formula were tested for the above-mentioned mechanical properties and the median value was used. For tensile properties, the stress–strain curves corresponding to the median value of tensile strength were illustrated.

#### 2.2.4. Hydrophobicity and its Transfer Characteristics 

In polluted regions, good hydrophobicity is desired to prevent pollution flashover accidents. Referring to IEC 61109:2008, the static contact angle was tested to evaluate the hydrophobicity and its transfer characteristics [27]. The dimension of the specimens is 120 mm × 50 mm × 6 mm. The sample surface was wiped with absolute ethanol to make sure that it was clean. Then, samples were placed in a dustproof container for 24 h at a temperature of 20 ± 5 °C. After that, the sample surface was contaminated with NSSD of 0.5 mg/cm^2^ (diatomite) and ESSD of 0.1 mg/cm^2^ (NaCl) using the quantitative brushing method. For each formula, three contaminated samples were prepared and the static contact angles of five points were measured for each sample. The final static contact angle is calculated by the average of 15 samples.

#### 2.2.5. Fourier Transform Infrared Spectroscopy

The internal functional groups in silicone rubber can be analyzed by FTIR due to its detection depth from several micrometers to tens of micrometers. In this paper, a Tensor 27 Spectrometer (Bruker Optics, Germany) was used to measure the characteristic peaks of silicone rubber in the spectral range from 4000 to 500 cm^−1^. The FTIR data were recorded in attenuated total reflection mode.

#### 2.2.6. X-ray Photoelectron Spectroscopy 

The material composition of the sample surface was analyzed by XPS with detection depth ranging from several nanometers to tens of nanometers [28]. In this study, the elemental composition and chemical state of the alicyclic epoxy resin surface and HTV silicone rubber surface subjected to UV-A radiation were analyzed by XPS (PHI Quantro SXM, ULVAC–PHI Co., Japan). The incidence angle and receiving angle were 54.7° and 90°, respectively. 

## 3. Results and Discussion

### 3.1. Appearance Analysis and SEM Studies 

Figure 3 shows the visual changes of alicyclic epoxy resin and silicone rubber materials with different UV aging times. The aging times for the samples are 0, 120, 240, 600, and 1000 h from left to right. With the increase of UV-A aging, white spots are observed on the surface of both samples. Thus, the surface color can reflect the aging time of both materials to some degree. Nevertheless, both insulating materials did not experience severe and obvious damage on the surface color during the experiment time. To further compare the variations in appearance with color changes, the average grave levels of the appearance pictures after different aging times are also calculated as shown in Figure 3. The calculated results illustrate that the average gray levels of both materials decrease with aging time. From the perspective of the average gray level, HTVSR experienced a more obvious change.

The appearance changes were imaged by SEM at 500× magnifications as shown in Figure 4. With the increase of UV-A aging time, holes and cracks of different sizes gradually appeared on the surfaces of both materials, and the surface roughness gradually increased. The reason for this observation is that some chemical bonds with weak bond energy in alicyclic epoxy resin and silicone rubber materials break. This leads to the destruction of macromolecular network structure and degradation of materials to a certain extent and the increase of surface defects. At the same time, the precipitation of Al(OH)_3_ and other fillers can also be observed due to the damage of matrix materials.

Figure 5 shows the cross-section of alicyclic epoxy resin and HTV silicone rubber after 1200 h UV-A radiation. From the microstructures of the cross-section images at 500× magnification, it can be observed that both samples have a relatively loose texture on the margin, and the interface between the matrix and fillers is obvious. This is ascribed to the UV-A radiation on the sample surface. It can lead to the breakage of the chemical bonds and the destruction of the macromolecular network structure. However, the microstructures in the inner side of the sample are not affected by the UV-A radiation.

### 3.2. Mechanical Characterization with UV-A Aging Time

Tensile and elongation at break of both samples are shown in Figure 6 and Figure 7. It can be observed that as the aging time increases, the tensile strength of alicyclic epoxy resin is enhanced. Conversely, the tensile strain property becomes inferior to the specimen without aging though it increases after 120 h UV-A aging time. In terms of HTVSR, with the increase of aging time, both properties are slightly lower than the sample without UV-A radiation.

When the aging time is 1000 h, the elongation at break of HTVSR material and alicyclic epoxy resin decreases by 32.89% and 13.63%, respectively. Moreover, the tensile strength of silicone rubber material decreases by 13.63% after 1000 aging as shown in Figure 6. On the contrary, the tensile strength of alicyclic epoxy resin increases by 22.58% after the 1000 h aging test.

The peak strain energy density was measured to analyze the mechanical properties of the two materials. It is calculated by the area surrounded by the material stress–strain curve. The physical significance of peak strain energy density is the mechanical energy consumed when the material is stretched to fracture stress per unit volume, which can be used to evaluate the toughness and impact resistance of the material.

With the increase of UV-A aging time, the peak strain energy density of HTVSR decreases gradually from 429 J∙m^3^ to 184 J∙m^3^. However, the peak strain energy density of alicyclic epoxy resin shows a fluctuating trend as shown in Figure 8. It increases slightly at first and then decreases after 100 h aging. Overall, the effect of UV-A radiation on the mechanical properties of HTVSR materials is more significant than it is on alicyclic epoxy resin. From the point of view of mechanical properties, alicyclic epoxy resin shows better UV aging resistance.

### 3.3. Hydrophobicity Analysis with UV-A Aging Time 

Without UV-A radiation, the static contact angle of alicyclic epoxy resin is 109°, while that of the silicone rubber material is 115°, as shown in Figure 9. The static contact angle of the silicone rubber material is larger than that of the alicyclic epoxy material, indicating that silicone rubber material has better hydrophobicity properties. This can be explained by the fact that the silicone rubber material is wrapped with methyl groups on both sides of the molecular main chain and the methyl groups have strong hydrophobicity. However, the alicyclic epoxy resin material contains more polar groups. This makes it inferior to silicone rubber in terms of hydrophobicity. 

Generally, the static contact angles of both materials decrease with the increasing UV-A aging time, and it is about 5° of decrease after 1000 h radiation. During the UV-A aging process, part of the groups on the surface of two materials are oxidized and polar molecules are generated. At the same time, the Si–C bond of the side chain of HTV silicone rubber material is also cut off, and the methyl group is reduced. All these reasons contribute to the increase of the molecular polarity of both materials and the decrease of hydrophobicity.

### 3.4. FTIR Analysis of Samples with UV-A Aging Time

FTIR analysis was conducted to illustrate the surface change of chemical structure and composition of alicyclic epoxy and silicone rubber materials which are the decisive factors of their macroscopic properties after being subjected to UV-A radiation. The main functional groups and characteristic absorption peaks of the two materials are shown in Table 1 and Table 2. Figure 10 and Figure 11 show Fourier transform infrared spectra of alicyclic epoxy resin and silicone rubber materials respectively after different UV-A aging times.

As shown in Figure 9, for alicyclic epoxy resin samples, the content of –OH near 3700–3200 cm^−1^ increases gradually as UV-A aging time increases, which indicates that a growing body of groups carry out oxidation reaction. Meanwhile, the content of C–H near 2970–2920 cm^−1^ and C–O–C near 1081 cm^−1^ gradually decreases. Furthermore, the content of C=O near 1770–1680 cm^−1^ shows a fluctuating trend as it increases first and then decreases.

As for HTV silicone rubber, with UV-A aging time increasing, the contents of –OH and Si–O–Si near 3700–3200 cm^−1^ and 1100–1000 cm^−1^ increase gradually while the contents of Si–C near 840–790 cm^−1^ decrease. After UV-A radiation, the characteristic peaks of Si–O–Si shifted to a higher wavenumber. That is relevant to the oxidation reaction. To be more specific, the methyl group on the side chain is cut off and the Si–O–Si is built due to the crosslinking reaction.

### 3.5. XPS Analysis of Samples with UV-A Aging Time

Figure 12, Figure 13, Figure 14 and Figure 15 show the effects of UV-A radiation on the surfaces of alicyclic epoxy resin and HTV silicone rubber with XPS analysis. In both kinds of materials, the Si, C, and O are dominant elements and their contents and position of binding energy were investigated. Based on the corrected peak area and sensitivity factor of each element, the relative percentage contents of Si, O, and C elements were calculated by normalization and the results are illustrated in Table 3.

After 1000 h UV-A aging, the C–O bond content of alicyclic epoxy resin increases from 3.08% to 7.90% and from 0% to 1.75% for the silicone rubber material. It indicates that both materials are oxidized in the UV-A aging process, resulting in the increase of O element content. 

As Table 3 shows, for alicyclic epoxy resin, the peak binding energy of Si, C, and O shifts to a lower binding energy region after 1000 h UV-A radiation. In contrast, HTVSR shows the opposite direction. Based on the chemical shifting theory, the decrease of the binding energy is induced by the combination with lower electronegativity atoms, which leads to the increase of electron density in the outer layer, and consequently the shielding effect is enhanced. For the alicyclic epoxy resin, the binding energy positions of the three main elements shift to a lower binding energy region, indicating that they bonded with more elements with low electronegativity after UV-A aging. Another reason is the loss of filler silica on the surface (tens of nanometers, the detection depths of XPS instruments). As for HTVSR, UV-A radiation cleaves the methyl group on the side chain and activates the crosslinking reaction, leading to more electronegative oxygen atoms bonded with silicon atoms.

### 3.6. Possible UV-A Degradation Mechanism with UV-A Aging

The macroscopic and microscopic properties of alicyclic epoxy resin and HTV silicone rubber materials are decreased by UV-A aging, and complex bond breaking, crosslinking, and oxidation reactions occur on the surface of materials, as illustrated in Figure 16 and Figure 17. For alicyclic epoxy resin, the reaction as shown in Figure 16 may occur when it suffers from UV-A radiation. The C–H bond in the alicyclic epoxy resin is oxidized, which increases the content of O element, C–O bonds, and C=O bonds in the molecule, while decreasing the content of C–H bonds. At the same time, under the combined actions of UV light and water, the ester bond will undergo a hydrolysis reaction, which increases the content of –OH in the alicyclic epoxy resin molecule. In addition, the bond energies of C–C and C–O are 332 kJ/mol and 326 kJ/mol, both are less than the energy of the UV-A ultraviolet photon which is 352 kJ/mol. Thus, C–C and C–O–C bonds in alicyclic epoxy resin molecules fracture to a certain extent, resulting in material deterioration.

## 4. Conclusions

In this study, the surface appearances, mechanical properties, and hydrophobicity of alicyclic epoxy resin and HTV silicone rubber subjected to different UV-A aging times are investigated and compared. It is found that the visual color of alicyclic epoxy resin changes obviously and a certain degree of fading phenomenon occurs after UV-A 1000 h aging test. However, the color of HTV silicone rubber is almost the same. In terms of mechanical properties, alicyclic epoxy resin behaves better than HTV silicone rubber after the UV-A aging test. As for alicyclic epoxy resin, tensile strength increases, and the elongation at break decreases. From the XPS and FTIR results, it is found that the interactions between fillers and the matrix become loose in both materials whereas the degree of crosslinking in the HTV matrix increases after UV-A aging. As the aging time increases, the hydrophobicity of both alicyclic epoxy resin and HTV silicone rubber decreases. HTV silicone rubber material shows a better UV-A aging resistance than alicyclic epoxy resin material. UV-A aging results in the oxidation of C–H in alicyclic epoxy resin and silicone rubber materials, and an increase in O element and C–O content. Meanwhile, C–O–C and C–C bonds in alicyclic epoxy resin materials and Si–C bonds in silicone rubber materials are cleaved, which led to the destruction of the macromolecular network structure and the degradation of materials to a certain degree. This resulted in an increase in surface defects.

## Figures and Tables

**Figure 1 polymers-14-04889-f001:**
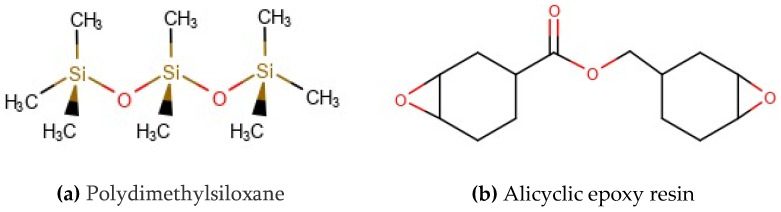
The chemical structure of materials: (**a**) HTVSR; (**b**) alicyclic epoxy resin.

**Figure 2 polymers-14-04889-f002:**
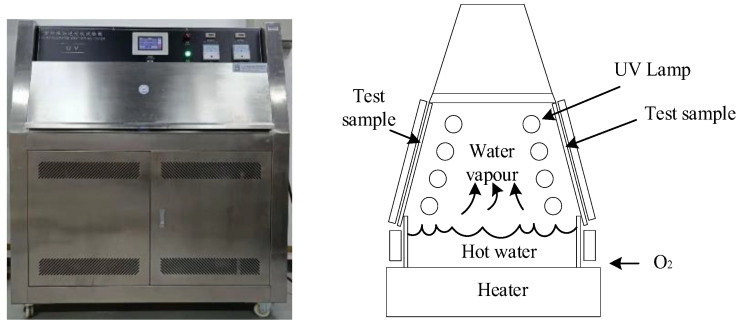
The UV aging test chamber.

**Figure 3 polymers-14-04889-f003:**
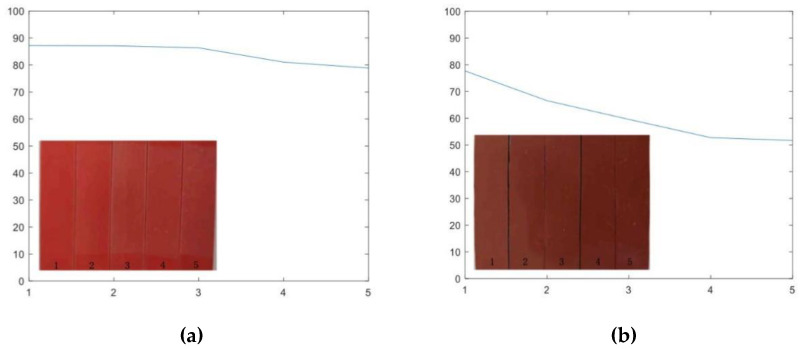
The appearance changes of two samples after different aging times: (**a**) The average gray level of the alicyclic epoxy resin sample; (**b**) the average gray level of the HTVSR sample.

**Figure 4 polymers-14-04889-f004:**
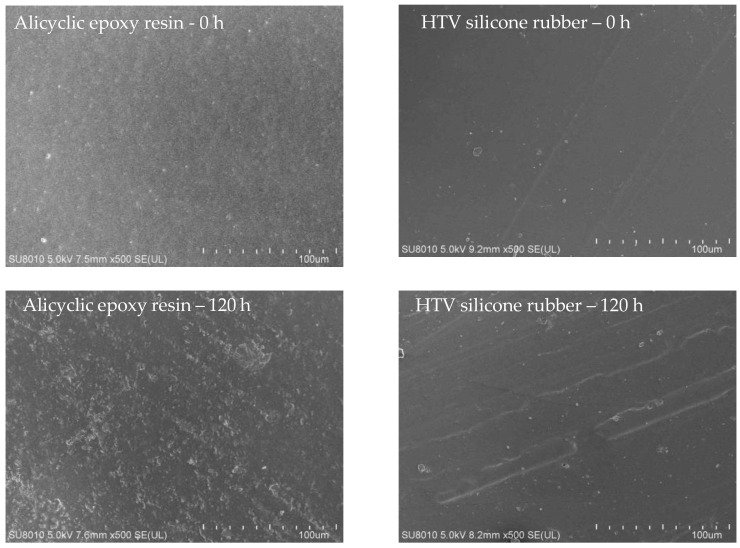
SEM of two samples with different UV-A aging times.

**Figure 5 polymers-14-04889-f005:**
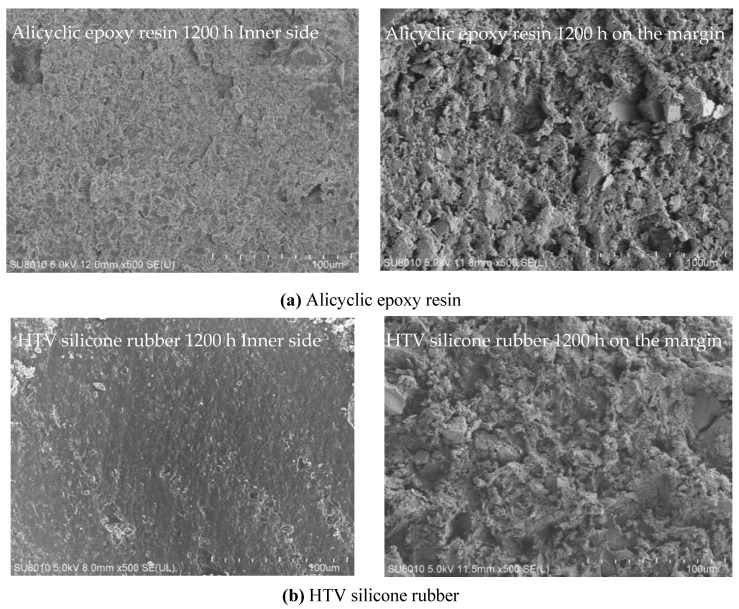
Cross-section images after 1200 h UV-A radiation: (**a**) Alicyclic epoxy resin; (**b**) HTV silicone rubber.

**Figure 6 polymers-14-04889-f006:**
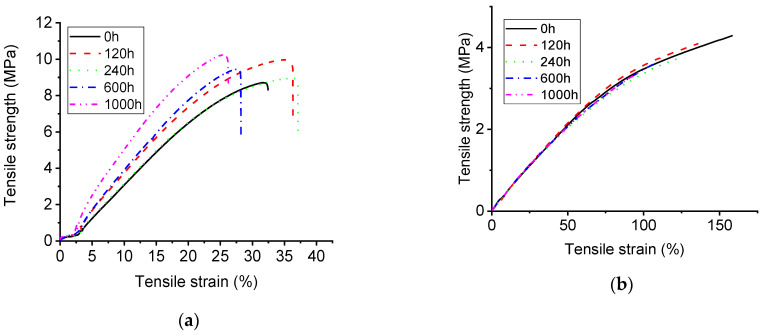
Tensile and elongation at break of two samples: (**a**) alicyclic epoxy resin; (**b**) HTVSR.

**Figure 7 polymers-14-04889-f007:**
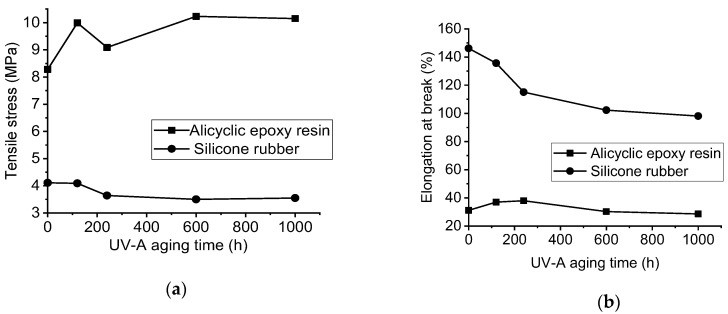
Comparison of mechanical properties of two samples at different UV-A aging time: (**a**) tensile stress; (**b**) elongation at break.

**Figure 8 polymers-14-04889-f008:**
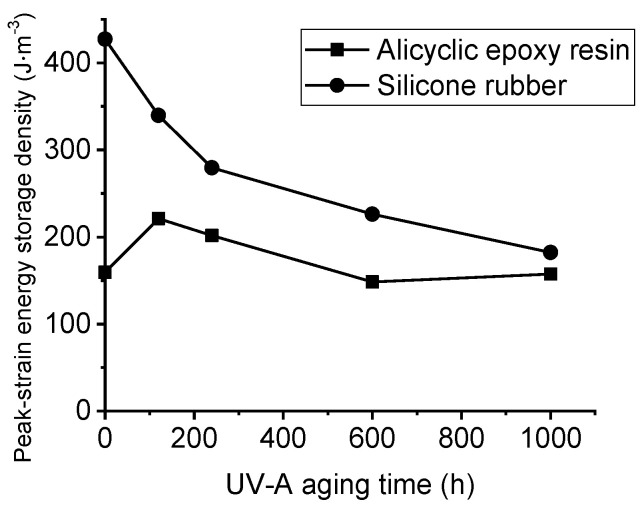
Comparison of the peak strain energy density of two samples at different UV-A aging time.

**Figure 9 polymers-14-04889-f009:**
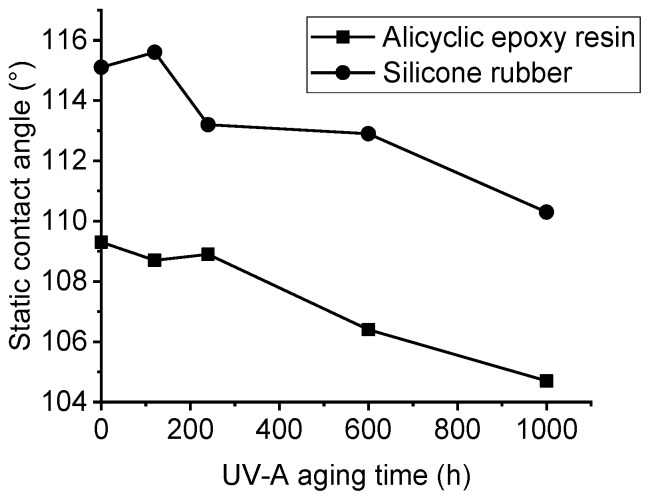
Static contact angle at different UV-A aging times of two samples.

**Figure 10 polymers-14-04889-f010:**
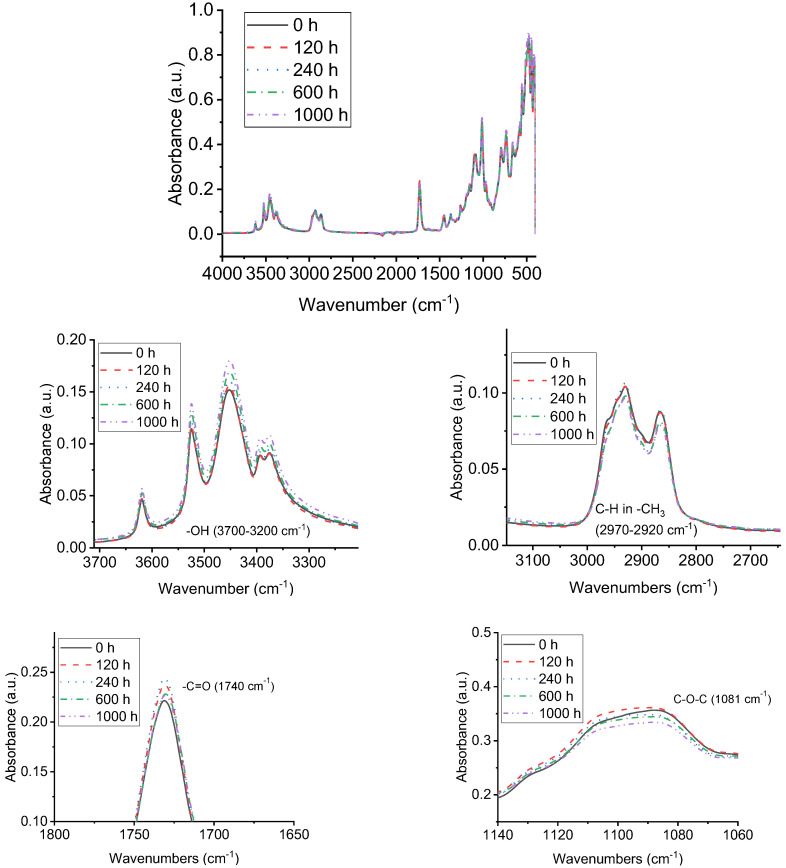
FTIR analysis of alicyclic epoxy resin samples at different UV-A aging times.

**Figure 11 polymers-14-04889-f011:**
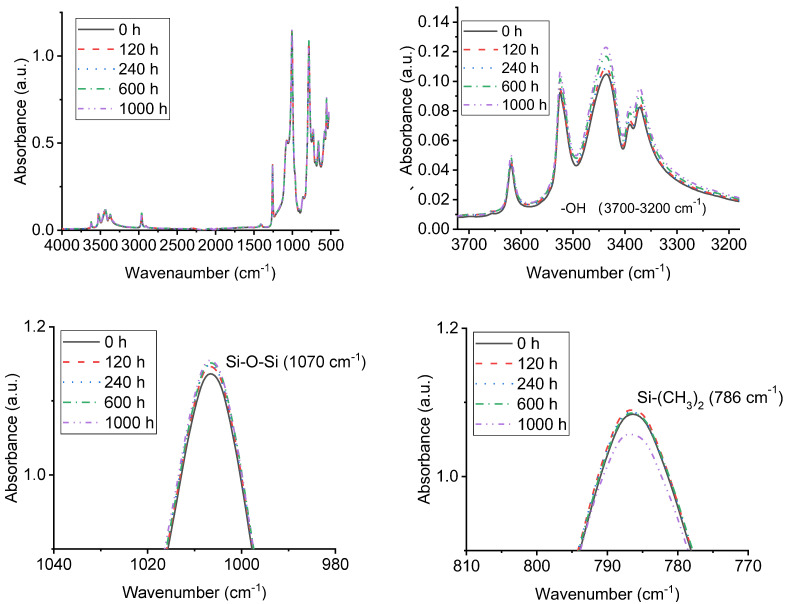
FTIR analysis of HTV silicone rubber at different UV-A aging times.

**Figure 12 polymers-14-04889-f012:**
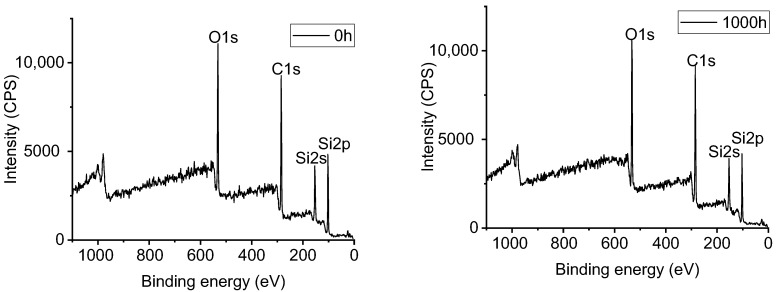
The XPS result of alicyclic epoxy resin samples.

**Figure 13 polymers-14-04889-f013:**
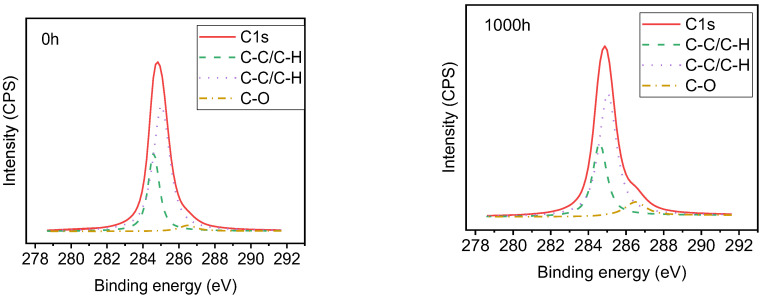
Peak fitting of C in alicyclic epoxy resin samples.

**Figure 14 polymers-14-04889-f014:**
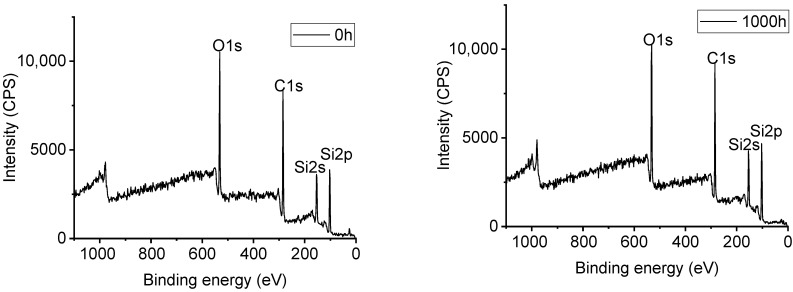
The XPS result of HTV silicone rubber samples.

**Figure 15 polymers-14-04889-f015:**
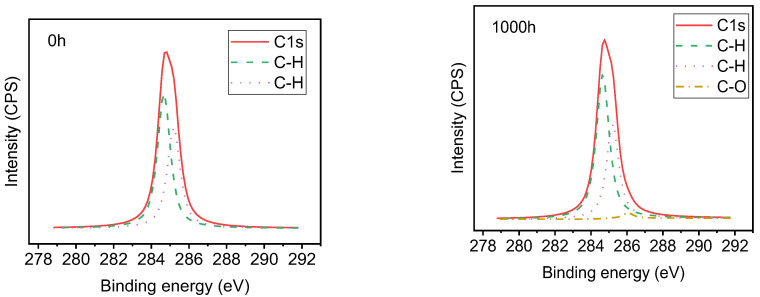
Peak fitting of C in HTV silicone rubber samples.

**Figure 16 polymers-14-04889-f016:**
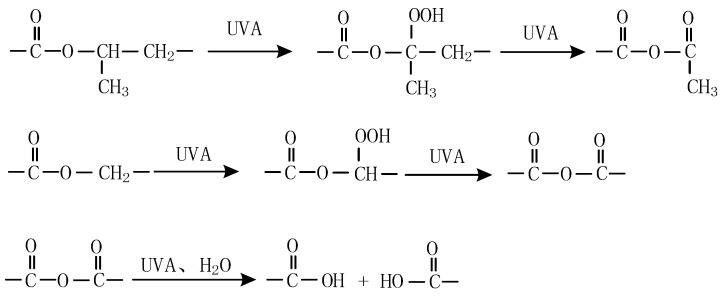
Proposed reactions of alicyclic epoxy resin during UV aging.

**Figure 17 polymers-14-04889-f017:**
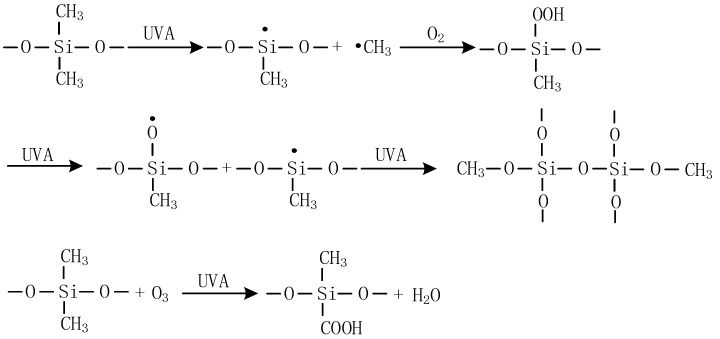
Proposed reactions of HTV silicone rubber during UV aging.

**Table 1 polymers-14-04889-t001:** The wavenumbers corresponding to the main chemical bond in alicyclic epoxy resin.

Chemical Bond	Wavenumbers (cm^−1^)
–OH	3700–3200
C–H (in –CH_3_)	2970–2920
C=O	1770–1680
C–O–C	1081

**Table 2 polymers-14-04889-t002:** The wavenumbers corresponding to the main chemical bond in HTV silicone rubber.

Chemical Bond	Wavenumbers (cm^−1^)
–OH	3700–3200
Si–O (in Si–O–Si)	1100–1000
Si–(CH_3_)_2_	840–790

**Table 3 polymers-14-04889-t003:** Comparison of the position of peak binding energy and contents of three elements of both samples after 1000 h UV-A aging time.

Sample	Aging Time(h)	Position of Peak Binding Energy (eV)			Content of Element (%)			
		Si	C	O	Si	C	O	C/O
Alicyclic epoxy resin	0	102.70	285.10	533.10	19.8	54.9	25.3	2.17
	1000	102.62	285.02	531.42	18.9	55.0	26.1	2.12
HTV silicone rubber	0	101.16	285.16	531.56	21.5	53.2	25.3	2.10
	1000	101.22	285.22	531.62	23.3	50.2	26.5	1.89

## Data Availability

Data presented in this study are available on request from the corresponding author.

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
