# Peer review of "Effect of Ultraviolet—A Radiation on Alicyclic Epoxy Resin and Silicone Rubber Used for Insulators"

_polymers, 2022, doi:10.3390/polym14224889_

Round 1

Reviewer 1 Report

In this manuscript, the authors studied the effect of UV-A on the high-temperature vulcanized silicone rubber (HTVSR) insulator to compare it with the alicyclic epoxy resin insulator. They used many techniques such as Scanning Electron Microscope (SEM), X-ray photoelectron spectroscopy (XPS), and Fourier transform infrared spectroscopy (FTIR) for studying the surface of two kinds of samples as a function of the UV-A aging time. Also, they analyzed the operation properties (mechanical properties, hydrophobicity) for outdoor insulators. They found that the influence of UV-A radiation is significant for alicyclic epoxy resin insulators. The mechanical properties of the alicyclic epoxy resin are maintained well and, the hydrophobicity decreases gradually. For silicone rubber, the mechanical properties of silicone rubber are decreased greatly. Both materials have broken chemical bonds, but the degree is relatively light, which meets the requirements of insulators for outdoor operation. This study is good and important to provide a promising strategy for producing outdoor insulators materials which are often subjected to high-intensity of ultraviolet radiation and can be used in coastal areas. The interpretations of the results are well discussed. The quantity and quality of the figures are appropriate. We believe that this research subject is promising for developing and studying insulator materials.

Summary: I recommend publishing this manuscript after considering my comments on the attached file.

Author Response

Reviewer 1:

In this manuscript, the authors studied the effect of UV-A on the high-temperature vulcanized silicone rubber (HTVSR) insulator to compare it with the alicyclic epoxy resin insulator. They used many techniques such as Scanning Electron Microscope (SEM), X-ray photoelectron spectroscopy (XPS), and Fourier transform infrared spectroscopy (FTIR) for studying the surface of two kinds of samples as a function of the UV-A aging time. Also, they analyzed the operation properties (mechanical properties, hydrophobicity) for outdoor insulators. They found that the influence of UV-A radiation is significant for alicyclic epoxy resin insulators. The mechanical properties of the alicyclic epoxy resin are maintained well and, the hydrophobicity decreases gradually. For silicone rubber, the mechanical properties of silicone rubber are decreased greatly. Both materials have broken chemical bonds, but the degree is relatively light, which meets the requirements of insulators for outdoor operation. This study is good and important to provide a promising strategy for producing outdoor insulators materials which are often subjected to high-intensity of ultraviolet radiation and can be used in coastal areas. The interpretations of the results are well discussed. The quantity and quality of the figures are appropriate. We believe that this research subject is promising for developing and studying insulator materials.

Summary: I recommend publishing this manuscript after considering my comments on the attached file.

Comments on PDF:

Q: The authors should add references or type the formula and process.

A: The alicyclic epoxy resin insulator studied in this paper was developed by our research group. The formula and process can be found in the following references and they are added to the revised manuscript.

  1. Liu Y, Lin Y, Wu K, et al. Analysis and Optimization on Non-uniformity of Temperature Distribution in Hydrophobic Cycloaliphatic Epoxy Resin Insulators during the Curing Process[J]. IEEE Transactions on Dielectrics and Electrical Insulation, 2021, 28(5): 1810-1818.
  2. Liu Y, Lin Y, Wang Y, et al. Simultaneously improving toughness and hydrophobic properties of cycloaliphatic epoxy resin through silicone prepolymer[J]. Journal of Applied Polymer Science, 2022: e52478.
  3. Liu Y, Lin Y, Cao B, et al. Enhancement of polysiloxane/epoxy resin compatibility through an electrostatic and van der Waals potential design strategy[J]. Polymer Testing, 2022: 107820.

Q: How did the author estimate 4 years?

A: Taking Guangzhou as an example, the annual natural solar radiation in Guangzhou is about 4234.62 MJ/m2. About 1% of ultraviolet light can reach the ground. The wavelength distribution of Type 1A (UVA-340) fluorescent ultraviolet lamps in GB/T 16422.3-2014 Plastics Laboratory Light Source Exposure Test Methods Part 3 Fluorescent Ultraviolet Lamps is 290 ≥ λ ≥ 320 accounting for 5.4%, 320 > λ ≥ 360 accounts for 38.2%, 360 > λ ≥ 400 accounts for 56.4%. According to the above data, it can be calculated that 1000 h aging under the UV aging conditions set in this paper is equivalent to 4 years of natural UV radiation in Guangdong.

Q: This decrease in the tensile strength for HTVSR is not clear in figure 4b, why? or do you mean the behavior in Fig. 5a, where the tensile strength decrease (not increase) ...?

A: We are sorry for not denoting the figure clearly and we have revised it in the paper. This decrease in the tensile strength is shown in figure 6a in the revised manuscript.

Q: This is difficult to see white color on the epoxy resin and also there is a change on the color for HTVSR?

A: We are sorry for the not clear description. There are white spots on the surface of both samples. The relevant description in the paper has been revised from line 154 to line 157.

Q: C-H dotes are not clear in Fig. 13.

A: The figure has been updated in the revised manuscript according to the comment.

Q: Reply to the comment “did the author measure the degree of crosslinking?” In conclusion

A: We have not tested the crosslinking density. Nevertheless, from the experiment data of FTIR, the characteristic peaks of Si-O-Si shifted to a higher wavenumber after UV-A radiation. That verified that the methyl group on the side chain is cut off and the Si-O-Si is built due to the crosslinking reaction.

Q: The title of the thesis “Watts, J. F. Master Thesis, Cambridge University Press,1998. ”

A: Sorry for the mistake. It should be “Shi, Q., Master Thesis, Electric Power University, 2014.” and it has been corrected in the revised manuscript.

Reviewer 2 Report

This manuscript entitled "Effect of ultraviolet-A radiation on acrylic epoxy resin and silicone rubber used for insulators" is interesting and well-organized.  It is suitable to publish in Polymers after minor revision.  Comments are given as follows:

1. The radiation distance between the radiation source and as-prepared materials should be provided.

2. Please provide and discuss the cross-section image of SEM observation. for the as-prepared materials. 

Author Response

Dear Editors and Reviewers:

Thank you very much for your comments on our manuscript entitled “The Effect of ultraviolet-A radiation on alicyclic epoxy resin and silicone rubber used for insulators”. These comments are valuable and helpful for revising and improving our manuscript. The responses to the reviewer’s comments are as follows:

Reviewer 2:

This manuscript entitled "Effect of ultraviolet-A radiation on acrylic epoxy resin and silicone rubber used for insulators" is interesting and well-organized. It is suitable to publish in Polymers after minor revision.  Comments are given as follows:

Q: 1. The radiation distance between the radiation source and as-prepared materials should be provided.

A: Thank you very much for your comments. The radiation distance between the radiation source and the specimens is 20 cm and it has been added into the revised manuscript.

  1. Please provide and discuss the cross-section image of SEM observation for the materials.

A:

Alicyclic epoxy resin 1200 h Inner side

Alicyclic epoxy resin 1200 h on the margin

(a) Alicyclic epoxy resin

HTV silicone rubber 1200 h Inner side

HTV silicone rubber 1200 h on the margin

(b) HTV silicone rubber

The cross-section images of alicyclic epoxy resin and HTV silicone rubber after 1200 h UV-A radiation.

From the microstructures of the cross-section images at 500X magnification, it can be observed that both samples have a relatively loose texture on the margin, and the interface between the matrix and fillers is obvious. This is ascribed to the UV-A radiation on the sample surface. It can lead to the breakage of the chemical bonds and the destruction of the macromolecular network structure. However, the microstructures in the inner side of the sample are not affected by the UV-A radiation.
